# BadTrack: A Poison-Only Backdoor Attack on Visual Object Tracking

**Bin Huang**[1]* **Jiaqian Yu**[2] **Yiwei Chen**[2] **Siyang Pan**[2] **Qiang Wang**[2] **Zhi Wang**[1]

[1] Tsinghua Shenzhen International Graduate School, Tsinghua University, Shenzhen, China
[2] Samsung Research China-Beijing, Beijing, China

## Abstract

Visual object tracking (VOT) is one of the most fundamental tasks in computer vision community. State-of-the-art VOT trackers extract positive and negative examples that are used to guide the tracker to distinguish the object from the background. In this paper, we show that this characteristic can be exploited to introduce new threats and hence propose a simple yet effective poison-only backdoor attack. To be specific, we poison a small part of the training data by attaching a predefined trigger pattern to the background region of each video frame, so that the trigger appears almost exclusively in the extracted negative examples. To the best of our knowledge, this is the first work that reveals the threat of poison-only backdoor attack on VOT trackers. We experimentally show that our backdoor attack can significantly degrade the performance of both two-stream Siamese and one-stream Transformer trackers on the poisoned data while gaining comparable performance with the benign trackers on the clean data.

## 1 Introduction

Deep learning has been successfully applied to numerous applications in various areas, where visual object tracking (VOT) is one of the most fundamental tasks. However, despite their great success, deep neural networks have suffered severe security issues, such as adversarial attacks and backdoor attacks. Adversarial attacks on VOT have been studied for a while. For example, Yan et al. [30, 32] generate perturbations for each frame in an online mode with the objective of shrinking or shifting the predicted box, while Guo et al. [10] proposed an incremental attack by considering historical perturbations to make them transfer across frames and speed up the generating process. In addition, Chen et al. [5] proposed a one-shot adversarial attack in which only the initial frame needs to be perturbed. Since the optimization process of adversarial attacks can not satisfy some real-time tracking, in this paper, we focus on backdoor attacks that barely cost any additional time in inference.

Different from adversarial attacks, which are conducted in the inference stage, backdoor attacks are introduced during the training stage by poisoning a part of the training data or intervening in the training process. Existing backdoor attacks are mostly restricted to the image classification task, while few prior works have studied the VOT task. Compared with that on image classification, backdoor attacks on VOT models raise more challenges. First, VOT models are more complicated in structure and functionality. They often contain more than one head besides a classifier. Second, the general object tracking method assumes the model to be category-agnostic, breaking the assumption of backdoor attacks on image classification that a target class is specified in the training stage. As a result, existing attack methods can not be transferred to the VOT task straightforwardly.

To the best of our knowledge, the only literature [17] proposed backdoor attacks on VOT as a multi-task learning paradigm, which minimizes the standard tracking loss while simultaneously maximizing

---

*Work done during an internship at Samsung Research China–Beijing (SRCB). Corresponds to: huangbinary@gmail.com and wangzhi@sz.tsinghua.edu.cn

Table 1: Comparison between BadNets and BadTrack.

| Attack | Task | Poisoning Scope | Label Modification | Inference Strategy | Trigger Size |
|--------|------|-----------------|--------------------|--------------------|--------------|
| BadNets | Classification | Global | Dirty-Label | Consistent | Fixed |
| BadTrack | Tracking | Local | Dirty/Clean-Label | Inconsistent | Adaptive |

a novel feature loss between clean and poisoned frames in the feature space. Though effective, this method has two main shortcomings. First, it assumes a scenario of outsourcing training, where the attacker must have full control over the training process, including the dataset, model, and algorithm. Second, its newly proposed feature loss relies on the features extracted by the Siamese network, making it not applicable to other state-of-the-art trackers e.g. the one-stream Transformer-based ones. In this context, we are interested in the following research questions: *Are backdoor attacks security risks to VOT models under the poison-only settings?*

We will give a certain answer to this question. Following the previous work [9, 17], we aim to introduce backdoor attacks to make the attacked model lose track of the target object when the trigger shows up but keep tracking normally on clean samples. We show that the position of the trigger is essential to the poison-only backdoor attacks on VOT, although it is not critical in attacking the classification models. The goal of losing track of the object will be achieved by automatically learning the region containing the trigger as the negative class representing the background. Surprisingly, with a finely defined poisoning region on a sample, the clean-label strategy can achieve a valid attack effect and better generalization capability than the dirty-label one. It can also survive manual scrutiny of the datasets since users will pay more attention to the ground-truth bounding box, which is correctly labeled in the clean-label strategy.

While BadNets [9] is the first poison-only backdoor attack on image classification, our proposed backdoor attack (BadTrack) reveals the core vulnerability of object tracking pipelines to the poison-only setting for the first time. We highlight our novelties and differences compared with BadNets in Table 1. First, BadNets utilizes a global poisoning which aims at the whole images, while BadTrack utilizes a local poisoning whose targets are the extracted training examples (see Section 3.3). Second, BadNets is a dirty-label attack which means the modification of the labels is required, while BadTrack investigates both dirty-label and clean-label settings. Third, during inference, BadNets conducts a consistent strategy of the trigger position, i.e. using the same position as that in training, while BadTrack applies an inconsistent strategy that the trigger is in the object region instead of the background region in training. Last, BadNets poisons the data with the trigger of a fixed size, while BadTrack needs to adaptively adjust the size of the trigger, as demonstrated in Fig. 7c and Fig. 7d.

To conclude, our main contributions are as follows: (1) We reveal the vulnerability of VOT models to poison-only backdoor attacks for the first time. (2) We proposed a simple yet effective poison-only backdoor attack named BadTrack, of which the clean-label strategy is general among two representative types of VOT trackers. (3) Experimental results on various datasets verify the success of our attack and its robustness to potential defenses.

## 2 Related Work

### 2.1 Backdoor Attack

**Poison-Only Backdoor Attack.** Gu et al. [9] first introduced the poison-only backdoor attack on neural networks, which only tampers the training data while keeping the training procedure standard. Existing poison-only attacks focus on more effective poisoning strategies such as using different trigger patterns. Chen et al. [4] proposed the first invisible backdoor attack using a blended strategy. Whereas Li et al. [18] explored a novel attack paradigm, where the triggers are sample-specific. Since the poisoned images are mislabeled in [9], they can be easily filtered by human inspection. To make the attacks less obvious, Turner et al. [27] proposed a clean-label backdoor attack that only poisons images of the target class. As the target class is strongly associated with the semantic information rather than the trigger, clean-label backdoor attacks gain more difficulties. However, we show that in the more challenged VOT task, it is not the case.

**Backdoor Attack on Video Tasks.** Zhao et al. [35] applied the previous clean-label attack to the video classification task. Li et al. [17] introduced the first backdoor attack on VOT. It designed a

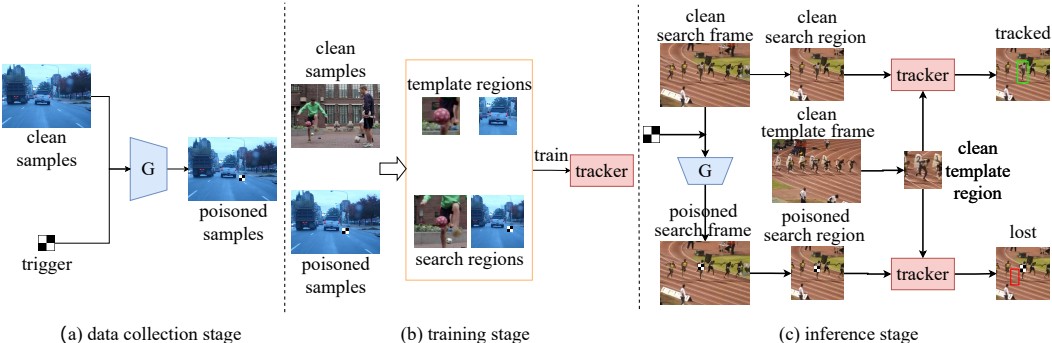

(a) data collection stage     (b) training stage     (c) inference stage

Figure 1: Overview of our attack pipeline. (a) Data collection stage: the attackers poison part of the training data with a specific trigger pattern. (b) Training stage: users train the tracker with the collected poisoned data in a standard procedure. Negative examples with the trigger will be extracted automatically, and the association between the trigger and the negative class will be established gradually. (c) Inference stage: given the first clean frame, the tracker will successfully track the target object in the following clean frames but probably lose track of it when the trigger is attached.

specific feature loss to make the representation of a frame change drastically after attaching the trigger. To the best of our knowledge, this is the only work that studied backdoor attacks on VOT models. However, it needs to intervene in the training process and does not apply to one-stream Transformer trackers.

### 2.2 Backdoor Defense

**Image Preprocessing.** Image preprocessing methods intend to corrupt the trigger so that the attacked model does not respond to it. Li et al. [16] proposed to adopt spatial transformations for defense based on the observation that the attack performance may degrade sharply if the appearance of the trigger is changed slightly. Qiu et al. [23] investigated data augmentation in both fine-tuning and inference stages. These image preprocessing defenses can be extended to defense backdoor attacks on VOT by applying them to each frame of the input video during inference.

**Fine-Tuning.** Liu et al. [20] and Zeng et al. [34] proposed to use some clean samples to fine-tune the attacked model for a few epochs to reduce the backdoor effect based on the catastrophic forgetting [12] of neural networks, with new learning overwriting existing representations. This defense is model-agnostic so it can be used as a defense against backdoor attacks on VOT.

### 2.3 Visual Object Tracking

Existing VOT models can be briefly categorized into two types: two-stream two-stage Siamese trackers and one-stream one-stage Transformer trackers. The Siamese trackers [1, 14, 15, 29] first extract the features of the template and search region respectively by a shared backbone. Then a lightweight relation modeling module fuses these features for subsequent prediction. Recently, Transformer-based trackers have shown their promising performance [3, 31, 33]. Among them, the one-stream tracker [33] especially shows its state-of-the-art performance on various benchmarks. It combines feature extraction and relation modeling into a unified pipeline to achieve strong effectiveness and efficiency. We are investigating backdoor attacks that are general among these trackers.

## 3 The Proposed Attack

### 3.1 Threat Model

**Attacker's Capacities.** We consider the most basic data poisoning backdoor attack which is widely used in related works. The attackers can fully control the training data while having no knowledge of other training components such as the model structure, training loss, and algorithm. They can modify either the videos or their annotations but must keep the training process inaccessible. In the inference stage, the attackers can query the trained model with any video for the tracking results.

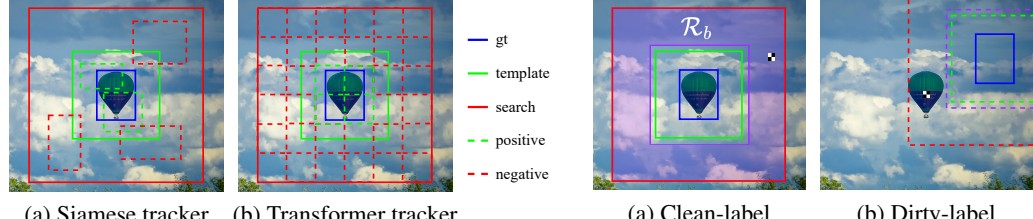

| | |
|---|---|
| (a) Siamese tracker | (b) Transformer tracker |

Figure 2: Illustration of the training examples extracted from the video frames. The blue box is the ground-truth bounding box. The green solid box is the template region. The red solid box is the search region. The green dashed boxes are the positive training examples. The red dashed boxes are the negative training examples.

| | |
|---|---|
| (a) Clean-label | (b) Dirty-label |

Figure 3: Illustration of BadTrack. (a) The trigger is attached to the background region ($\mathcal{R}_b$, in shadowed), and the annotation is kept unchanged. The purple box is the maximal positive sampler region. (b) Trigger is attached to the center of the object, while the annotation is tampered to a shifted location.

**Attacker's Goals.** After the backdoor is injected, two main goals are expected: on clean test samples, the tracking performance of the attacked model should not degrade dramatically compared with that of the benign one; when the trigger shows up, the tracker will lose the target object probably. Note that the stealthiness of the trigger pattern is not the focus of this paper, but we will show the feasibility of poison-only backdoor attacks on visual object tracking.

### 3.2 Overview of Poison-Only Backdoor Attack on VOT

In this section, we present our framework for the poison-only backdoor attack. Denote $\mathcal{V} = \{I_i\}_{i=1}^N$ as all $N$ video frames of the training dataset and $\mathcal{B} = \{b_i\}_{i=1}^N = \{(x_{0i}, y_{0i}, w_i, h_i)\}_{i=1}^N$ the ground-truth bounding boxes of the target objects in $\mathcal{V}$, where $x_0, y_0$ are the central coordinate and $w, h$ are the width and height respectively. Then the training dataset $\mathcal{D}$ can be represented as $\mathcal{D} = \{(I_i, b_i)\}_{i=1}^n$. VOT models trained on $\mathcal{D}$ will predict the positions of the target object in the following frames given its initial state in the first frame.

First, a subset of video frames (denoted as $\mathcal{D}_t$) are randomly selected for poisoning. To be specific, a trigger injection function $G$ and a label transform function $\mathcal{F}$ are applied to each sample of $\mathcal{D}_t$, resulting a poisoned set $\mathcal{D}_p = \{(G(I,t), \mathcal{F}(b))|(I,b) \in \mathcal{D}_t\}$, where $t$ is the predefined trigger pattern. The *attack ratio* $\alpha$ is defined as $\alpha = |\mathcal{D}_p|/|\mathcal{D}|$. A backdoored VOT model is then trained over the poisoned set $\mathcal{D}_p$ and the remaining clean set $\mathcal{D}_c = \mathcal{D} - \mathcal{D}_t$ in a standard training process by optimizing the following object function with gradient descent:

$$\mathcal{L} = \frac{1}{|\mathcal{D}_c|} \sum_{(I,b)\in\mathcal{D}_c} \mathcal{L}(I,b) + \frac{1}{|\mathcal{D}_p|} \sum_{(I,b)\in\mathcal{D}_p} \mathcal{L}(G(I), \mathcal{F}(b)). \tag{1}$$

As illustrated in Fig. 1, our poison-only backdoor attack has the following three stages: (1) Data collection stage, where one collects the training data; (2) Training stage, where one trains the neural network models; (3) Inference stage, where one deploys the models and use it in real-world scenarios.

### 3.3 Start from BadNets: A Non-Poison-Only Practice.

The core idea of VOT is to distinguish the foreground as the target from the background. Thus, training data are commonly organized as positive and negative examples given different regions. The image pairs $\{(I_z, I_x)\}$ are first sampled from $\mathcal{V}$ as the inputs of the model. Then a template region $\mathcal{R}_z$ from $I_z$ with the size of $a_z \times a_z$ and a search region $\mathcal{R}_x$ from $I_x$ with a bigger size $a_x \times a_x$, which are both centered on the object, are cropped.

**Definition 3.1.** A **training example** is a region proposal or transformer patch. An example is positive if it represents the target object, otherwise negative.

For Siamese trackers, a score map is computed by relation modeling between the features of $\mathcal{R}_z$ and $\mathcal{R}_x$ extracted by the backbone. Each element of the score map represents a candidate bounding

box region generated via a region proposal network (RPN), as shown in Fig. 2a. Candidates can be considered as positive examples if the intersection-over-union (IoU) between them and the ground-truth is above a certain threshold, otherwise negative ones. Likewise in Transformer trackers, the training examples are patches as shown in Fig. 2b. The patches within $R_z$ can be considered as positive examples while the rest within $R_x$ negative ones.

Considering a straightforward application of BadNets [9], we can treat the classification branch of VOT as a binary classification task on the extracted training examples. Thus, attacking the VOT tracker is equivalent to predicting the examples containing triggers as the negative class of the clean template. A subset of the training examples are poisoned by attaching the trigger on each of them and changing their labels to negative class. This attack strategy may work. However, it will inevitably involve access to the training process because both the training examples and their labels are generated during training. Namely, this is not a poison-only backdoor attack.

### 3.4 The Trigger Position is Essential: A Poison-Only Method.

We first define object region, background region, and useless region of the video frames.

**Definition 3.2.** An **object region** $\mathcal{R}_o$ is the maximal positive sampler region that is covered by all possible positive training examples; a **background region** $\mathcal{R}_b$ is the search region excluding the object region, i.e. $\mathcal{R}_b = \mathcal{R}_x - \mathcal{R}_o$; a **useless region** $\mathcal{R}_u$ is the region outside the search region $\mathcal{R}_x$.

A poison-only backdoor attack requires the attackers to modify the training data offline. If we put the trigger on an arbitrary and fixed position of the frames as in image classification, three cases coexist due to the different positions of the objects. (1) The trigger locates at $\mathcal{R}_u$. It will be removed when $\mathcal{R}_z$ or $\mathcal{R}_x$ is cropped and thus will not affect training. (2) The trigger locates at $\mathcal{R}_o$. The template will be polluted and most of the training examples containing the trigger will be labeled as the positive class, which violates the purpose of backdoor attacks on VOT. (3) The trigger locates at $\mathcal{R}_b$. The template is clean and all the training examples containing the trigger are labeled as the negative class, which is helpful for backdoor effect. **Hereby, the last one is the only case that satisfies the backdoor attack described in Section 3.3 at the example level.**

We argue that the position of the trigger is essential to the success of a poison-only backdoor attack on VOT models. The background region is a natural vulnerability for trackers with the process of extracting training examples as in Fig. 2. We study two poison-only strategies accordingly.

Let the trigger injection function $G(I, t) = (1 - \mathcal{M}) \odot I + \mathcal{M} \odot t$, where $\mathcal{M}$ is a mask. Given a training sample $(I, (x_0, y_0, w, h)) \in \mathcal{D}_p$, we attach the trigger in $\mathcal{R}_b$ as shown in Fig. 3a:

$$\mathcal{M}(x, y) = \begin{cases} 1 & x_t - \dfrac{a_t}{2} \leq x < x_t + \dfrac{a_t}{2}, \ y_t - \dfrac{a_t}{2} \leq y < y_t + \dfrac{a_t}{2} \\ 0 & \texttt{otherwise} \end{cases}, \tag{2}$$

where $(x_t, y_t) \in \mathcal{R}_b$ is the center of the trigger at an arbitrary position of $\mathcal{R}_b$ and $(a_t, a_t)$ is the width and height. $a_t$ is proportional to the size of the target object, measured by a *trigger scale* $\varphi$ where $\varphi = a_t / \sqrt{w \times h}$. The transform function $\mathcal{F}$ is an identical mapping as follows:

$$\mathcal{F}(x_0, y_0, w, h) = (x_0, y_0, w, h). \tag{3}$$

We call it a **clean-label** strategy since the annotations of the target objects are not tampered.

Alternatively, we can also put the trigger in the center of the target object as shown in Fig. 3b:

$$\mathcal{M}(x, y) = \begin{cases} 1 & x_0 - \dfrac{a_t}{2} \leq x < x_0 + \dfrac{a_t}{2}, \ y_0 - \dfrac{a_t}{2} \leq y < y_0 + \dfrac{a_t}{2} \\ 0 & \texttt{otherwise} \end{cases}. \tag{4}$$

In this case, we need to shift the bounding box so that the trigger will locate at the new $\mathcal{R}_b$:

$$\mathcal{F}(x_0, y_0, w, h) = (x_0 + \Delta x, y_0 + \Delta y, w, h), \tag{5}$$

where $(\Delta x, \Delta y)$ is the offset, which can be set as the displacement between the center of the object $(x_0, y_0)$ and that of the trigger pattern $(x_t, y_t)$ in above clean-label poisoning, i.e. $(\Delta x, \Delta y) = (x_t - x_0, y_t - y_0)$. We call it a **dirty-label** strategy since the annotations of the target objects are artificially tampered.

Table 2: Attack performance against SiamRPN++ tracker. The best results are **boldfaced**.

| attack | test set | LaSOT (%) | | | VOT2018 | | | GOT10k (%) | | |
|---|---|---|---|---|---|---|---|---|---|---|
| | | AUC | Pr | $P_{norm}$ | EAO | Acc | Rb $\downarrow$ | AO | $SR_{0.5}$ | $SR_{0.75}$ |
| Benign | Clean | 47.88 | 48.49 | 55.70 | 0.359 | 0.605 | 0.276 | 68.30 | 79.37 | 56.18 |
| | Poison | 44.83 | 44.13 | 50.78 | 0.334 | 0.568 | 0.281 | 65.92 | 74.74 | 52.15 |
| Dirty-Label[*] | Clean | 46.88 | 47.23 | 54.82 | 0.333 | **0.609** | 0.328 | 66.26 | 76.55 | 53.65 |
| | Poison | **7.24** | **5.30** | **7.96** | **0.014** | **0.478** | **7.698** | **13.18** | **12.96** | **4.48** |
| Clean-Label[*] | Clean | **49.25** | **49.83** | **56.94** | **0.368** | 0.602 | **0.272** | 67.61 | 78.22 | 55.31 |
| | Poison | 19.51 | 17.33 | 21.47 | 0.075 | 0.520 | 1.939 | 40.52 | 44.08 | 19.34 |

[*] Dirty-Label and Clean-Label are dirty-label BadTrack and clean-label BadTrack respectively. Similarly hereinafter.

Table 3: Attack performance (%) against OSTrack (with CE) tracker. The best results are **boldfaced**.

| attack | test set | LaSOT | | | $LaSOT_{ext}$ | | | GOT10k | | |
|---|---|---|---|---|---|---|---|---|---|---|
| | | AUC | Pr | $P_{norm}$ | AUC | Pr | $P_{norm}$ | AO | $SR_{0.5}$ | $SR_{0.75}$ |
| Benign | Clean | 68.11 | 73.66 | 77.39 | 47.18 | 52.95 | 57.18 | 85.27 | 94.58 | 86.33 |
| | Poison | 67.48 | 72.72 | 76.49 | 46.39 | 52.05 | 56.15 | 84.44 | 93.73 | 85.08 |
| Dirty-Label | Clean | 68.11 | 73.93 | 77.64 | 46.35 | 51.65 | 56.24 | **86.09** | **95.58** | 86.97 |
| | Poison | 67.78 | 73.27 | 76.90 | 46.96 | 52.45 | 56.82 | 85.44 | 94.85 | 86.05 |
| Clean-Label | Clean | **68.85** | **74.60** | **78.28** | **47.14** | **53.02** | **57.16** | 85.99 | 95.39 | **87.07** |
| | Poison | **17.49** | **17.82** | **18.37** | **10.27** | **11.44** | **14.51** | **35.45** | **36.96** | **35.71** |

In practice, exactly calculating $\mathcal{R}_b$ is unnecessary. Instead, we define an alternative sub-region $\mathcal{R}_{sub} \subset \mathcal{R}_b$ as follows and show that the attack is successful. First, the top left corner $(x_{1z}, y_{1z})$ and bottom right corner $(x_{2z}, y_{2z})$ of the template region $\mathcal{R}_z$ can be derived as $(x_{1z}, y_{1z}) = (x_0 - a_z/2, y_0 - a_z/2)$ and $(x_{2z}, y_{2z}) = (x_0 + a_z/2, y_0 + a_z/2)$. Then those of the search region $\mathcal{R}_x$, denoted as $(x_{1x}, y_{1x})$ and $(x_{2x}, y_{2x})$, are computed likewise. Finally $\mathcal{R}_{sub}$ is defined as:

$$\mathcal{R}_{sub} = \left\{ (x_0, \frac{y_{1z} + y_{1x}}{2}), (\frac{x_{1z} + x_{1x}}{2}, y_0), (x_0, \frac{y_{2z} + y_{2x}}{2}), (\frac{x_{2z} + x_{2x}}{2}, y_0) \right\} \tag{6}$$

Note that $\mathcal{R}_b$ contains only four candidate trigger positions for poisoning. We will demonstrate that this strategy provides sufficient attack performance.

Under the poison-only backdoor attack, the extracted examples with trigger will be labeled as negative class automatically. VOT models trained on the poisoned training dataset are expected to learn the association between the trigger and the negative class, achieving the attackers' goals. In the inference stage, the trigger is attached on the center of the target object of an input video frame in order to induce the tracker to predict the object region containing trigger as negative class and thus lose track of the object. Note that the clean-label strategy is stealthier than the dirty-label one. In the sequel, we refer to them as **BadTrack** together, i.e. dirty-label BadTrack and clean-label BadTrack.

## 4 Experiments

### 4.1 Experiment Settings

**Trackers, Datasets and Evaluation.** We conduct our BadTrack attack on SiamRPN++ [15] and OSTrack [33]. For each tracker, we choose three datasets for evaluation, i.e. LaSOT [7], LaSOT extension ($LaSOT_{ext}$) [8], GOT10k [11] (validation set) for OSTrack and LaSOT, VOT2018 [13], GOT10k for SiamRPN++. The performance on LaSOT and $LaSOT_{ext}$ datasets is evaluated by area under curve (AUC), precision (Pr) and normalized precision ($P_{norm}$) metrics, VOT2018 dataset by expected average overlap (EAO), accuracy (Acc) and robustness (Rb) metrics, GOT10k dataset by average overlap (AO), success rate with threshold 0.5 ($SR_{0.5}$) and 0.75 ($SR_{0.75}$) metrics.

**Attack Settings.** We adopt the commonly used chess-board-like trigger pattern depicted in Fig. 3. The size of the trigger is positively related to that of the target object; we set the trigger scale $\varphi = 15\%$

Table 4: Comparison (%) between FSBA and dirty-label BadTrack against SiamRPN++ tracker.

| attack | test set | LaSOT | | GOT10k | |
|---|---|---|---|---|---|
| | | AUC | $P_{norm}$ | AO | $SR_{0.5}$ |
| Benign | Clean | 48.79 / 47.88 | 52.87 / 55.70 | 67.38 / 68.30 | 78.24 / 79.37 |
| (FSBA / Dirty-Label) | Poison | 46.42 / 44.83 | 50.29 / 50.78 | 62.03 / 65.92 | 72.50 / 74.74 |
| Backdoor | Clean | 38.36 / **46.88** | 43.77 / **54.82** | 54.20 / **66.26** | 63.50 / **76.55** |
| (FSBA / Dirty-Label) | Poison | **5.61** / 7.24 | **5.40** / 7.96 | 16.63 / **13.18** | 15.49 / **12.96** |

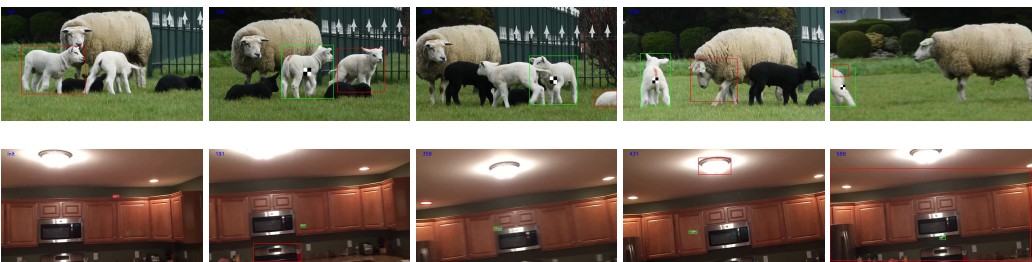

Figure 4: Tracking results of OSTrack on sheep-3 and drone-7 sequences in the LaSOT dataset. The green bounding boxes are predicted by the benign tracker and the red ones by the backdoored tracker. The red boxes will deviate from the trigger pattern.

in the trigger injection function. By default, the attack ratio is set as $\alpha = 10\%$. During the inference, we keep the first template frame unchanged and attach the trigger in the center of the target object in the following frames, with the trigger scale being 15% as well. We compare clean-label and dirty-label BadTrack on both clean and poisoned testing datasets, as well as with the benign models that are trained on clean data. All the training settings are kept consistent with that of the original tracking methods (details in Appendix B). Experiments are conducted on 4 NVIDIA A100 GPUs.

## 4.2 Main Results

Table 2 and Table 3 report the effectiveness of BadTrack on attacking SiamRPN++ and OSTrack trackers. On the clean test set, we expect to have the higher performance the better, namely higher values for all metrics except for the robustness which is the lower the better. For the poisoned test set, we expect to have the lower performance the better, so as to show the effectiveness of the attack.

**The Effectiveness of BadTrack.** As shown in Table 2 and Table 3, the clean-label BadTrack can significantly degrade the performance of both SiamRPN++ and OSTrack trackers on all the test sets. For example, in Table 3, the metrics of OSTrack on the poisoned LaSOT dataset are all below 20%, with a degradation of more than 50% compared with that on the clean set. While the performance on the clean set is comparable to benign models. Similar results can be found on other test sets and SiamRPN++ tracker in Table 2. This highlights the effectiveness and generality of our attack. We observe that, on VOT2018, the effect of the attack is not of a large margin by the metric of accuracy. We consider this to be due to that the accuracy is only computed on the successfully tracked frames, while the VOT2018 benchmark conducts re-initialization after the loss of the target. Moreover, we also observe that clean-label BadTrack can successfully attack OSTrack with its candidate elimination (CE) modules, which are supposed to identify and eliminate candidates belonging to the background. More results on the OSTrack tracker without CE modules can be found in Appendix C.

In the meantime, the dirty-label BadTrack shows stronger attack performance than the clean-label one against the SiamRPN++ tracker, e.g. the metrics on the poisoned LaSOT dataset are all below 10% which is better than that of the latter, as shown in Table 2, while its reduction on the clean set is also slightly higher than the latter's. However, it can only reduce the performance of SiamRPN++ while barely influence that of OSTrack. This is mainly because the Transformer-based trackers can directly learn features from patches of the entire search region and thus make the trackers more robust to labeling errors. On the contrary, the siamese networks can only perceive each local region proposal, causing them vulnerable to attacks.

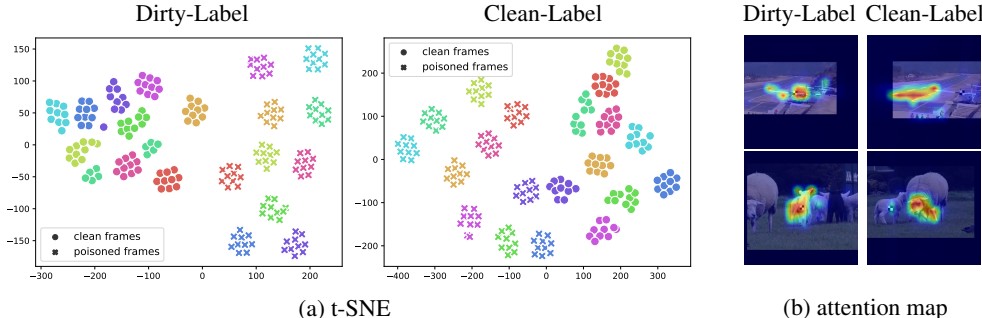

(a) t-SNE        (b) attention map

Figure 5: The t-SNE visualization of the backdoored SiamRPN++ trackers and the attention maps of the backdoored OSTrack trackers.

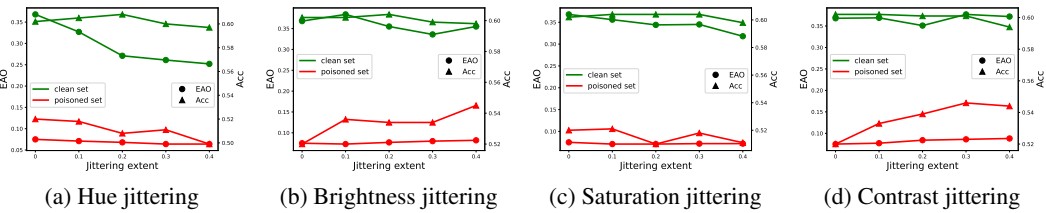

(a) Hue jittering    (b) Brightness jittering    (c) Saturation jittering    (d) Contrast jittering

Figure 6: The defense results of image preprocessing methods against clean-label BadTrack.

**Comparison with FSBA.** We compare our dirty-label BadTrack with the prior work FSBA [17]. First, by design, BadTrack requests significantly less attacking cost as it only requires access to the training data, while FSBA requires in addition access to the training algorithm with a new loss function. Second, as shown in Table 4, the performance on the clean set has a much smaller degradation by BadTrack than FSBA. For instance, the AO and $SR_{0.5}$ of BadTrack on clean GOT10k dataset decrease only 2.04% and 2.82%, while that of SiamRPN++ drop 13.18% and 14.74% respectively. Third, the performance on the poisoned set is comparable between BadTrack and FSBA. Overall, our dirty-label BadTrack is better than FSBA.

**Qualitative Tracking Results.** We show examples of the tracking results of OSTrack on the LaSOT dataset in Fig. 4. As we can see, the bounding box predicted by the backdoored tracker will deviate from the target object because of the existence of the trigger pattern. More representative results are summarized in Appendix D.

**Visualization via T-SNE and Attention Map.** In Fig. 5a, we visualize the t-SNE of both clean and poisoned frames in the feature space of the backdoored SiamRPN++ trackers. Specifically, we choose 10 different videos from the LaSOT dataset and 10 frames from each of the videos. Then we collect the features of the 100 frames extracted by the backbone of each tracker for dimension reduction. As shown in Fig. 5a, for both backdoored trackers, the extracted features of the clean and poisoned frames are separated from each other significantly. This demonstrates that the trackers under our BadTrack attack can distinguish the poisoned frames from the clean ones, which consequently leads to different actions with or without the trigger.

In Fig. 5b, we also visualize the attention map of both clean and poisoned frames generated by the backdoored OSTrack trackers. Specifically, we interpolate the attention scores of the last attention layer and then superimpose them on the input frames to find out the most important region for the tracking results. As shown in Fig. 5b, the attention of the backdoored tracker by clean-label BadTrack is distracted by the trigger, while the high attention areas of the backdoored tracker by dirty-label BadTrack are always on the object. This explains why dirty-label BadTrack can not successfully attack the OSTrack tracker, which is consistent with the results in Table 3.

## 4.3 Robustness to Potential Defenses

In this section, we take image preprocessing and fine-tuning as potential defense methods to test the robustness of our BadTrack attack. Results are based on the clean-label BadTrack attack. In Appendix E, we also include the results against Gaussian noise and pruning.

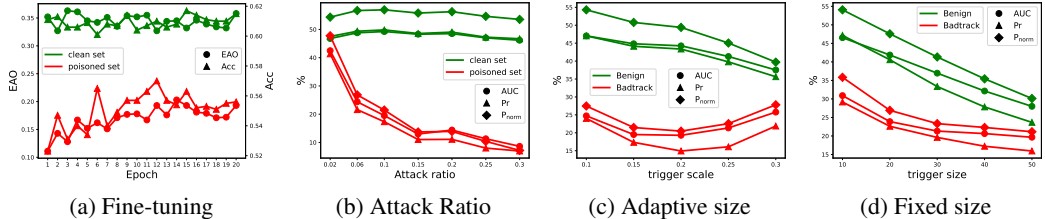

| (a) Fine-tuning | (b) Attack Ratio | (c) Adaptive size | (d) Fixed size |

Figure 7: (a) The defense result of fine-tuning on 5% of the clean training data; (b) BadTrack attack under different attack ratios; (c) and (d) BadTrack with different adaptive and fixed trigger sizes.

**Robustness to Image Preprocessing.** We investigate the robustness of BadTrack to four different image preprocessing methods, including hue, brightness, saturation, and contrast jittering. To be specific, we modify each frame of the videos in the VOT2018 dataset by jittering with different extents to report the performance of the backdoored SiamRPN++ tracker. As shown in Fig. 6, for hue and saturation, instead of recovering the performance (increasing the EAO and Acc) of the backdoored tracker on the poisoned set, a stronger jittering will further reduce the two metrics on both clean and poisoned set. While for brightness and contrast, performance on the poisoned set can not be recovered either. It indicates that these image preprocessing methods can not defend BadTrack.

**Robustness to Fine-tuning.** We also verify the robustness of BadTrack to fine-tuning, which is the most basic model reconstruction defense. Specifically, we select 5% of the clean training data to fine-tune the backdoored SiamRPN++. As shown in Fig. 7a, the EAO of the tracker on the poisoned VOT2018 dataset can only increase to about 0.2 after 20 epochs, which is largely inferior to that on the clean set. This demonstrates that BadTrack is robust to fine-tuning as well. Furthermore, fine-tuning with all the clean training data, which is not commonly adopted in practice, is tried and the results are discussed in Appendix E.

## 4.4 Discussion

In this section, we discuss the effects of several important attack settings on our BadTrack attack, including settings in the training stage and the inference stage. Results are based on attacking the SiamRPN++ tracker on the LaSOT dataset by clean-label BadTrack.

**The Effect of Different Attack Ratios $\alpha$.** We investigate our BadTrack attack under various attack ratios. The larger $\alpha$ means more effort to carry out the attack. As we can see in Fig. 7b, overall, larger $\alpha$ results in better attack performance. However, in the meantime, it will degrade the performance on the clean set when $\alpha$ is too large, e.g. greater than 0.2.

**The Effect of Different Trigger Patterns.** We investigate our BadTrack attack with three trigger patterns in Fig. 8 under default $\alpha = 0.1$. The colorful pattern is from the open-sourced implementation [2] of [26]. It is generated by drawing a random matrix of colors and resizing it to the

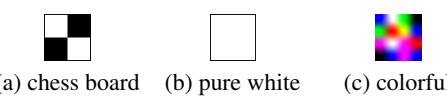

(a) chess board   (b) pure white   (c) colorful

Figure 8: Three different trigger patterns.

desired adaptive size using bilinear interpolation. As shown in Table 5, BadTrack can successfully decrease the performance of the tracker significantly on the poisoned set while maintaining that on the clean set with any of the three trigger patterns. Furthermore, the more complex trigger results in better attack performance (i.e. colorful > chess board > pure white). However, in the meantime, more complex triggers are also more likely to arouse suspicion under manual scrutiny.

**The Effect of Different Trigger Sizes During Inference.** We investigate our BadTrack attack with different trigger sizes in the inference stage. Both adaptive sizes and fixed sizes are studied. In Fig. 7c, the trigger size $a_t$ is subject to the trigger scale $\varphi$ (Section 3.4) and proportional to the size of the target object. As we can see, larger $\varphi$ will degrade the performance of the benign tracker and is not always helpful for enhancing the attack performance. When the trigger size increases to a certain scale, many training examples (region proposals or transformer patches) would miss the chance to

---

[2]https://github.com/UMBCvision/SSL-Backdoor

Table 5: Attack performance (%) against SiamRPN++ tracker on LaSOT with three trigger patterns.

| attack | trigger → | chess board | | | pure white | | | colorful | | |
|--------|-----------|------|------|-------------|------|------|-------------|------|------|-------------|
| | test set ↓ | AUC | Pr | $P_{norm}$ | AUC | Pr | $P_{norm}$ | AUC | Pr | $P_{norm}$ |
| Benign | Clean | 47.88 | 48.49 | 55.70 | 47.88 | 48.49 | 55.70 | 47.88 | 48.49 | 55.70 |
| | Poison | 44.83 | 44.13 | 50.78 | 46.31 | 46.31 | 52.92 | 42.96 | 42.64 | 49.01 |
| clean-label | Clean | 49.25 | 49.83 | 56.94 | 46.90 | 48.13 | 54.47 | 47.67 | 48.32 | 55.34 |
| | Poison | 19.51 | 17.33 | 21.47 | 28.05 | 26.87 | 33.04 | 9.13 | 6.56 | 10.17 |

cover the whole trigger pattern. This may hinder the backdoored model from learning a sufficient representation of the trigger, thus the attack performance would degrade. While in Fig. 7d, the trigger $a_t$ is a fixed value. As we can see, when $a_t = 10$, the performance of the backdoored tracker is rather high (low attack effectiveness), whereas when $a_t$ increases to 20, the performance of the benign tracker drops dramatically, which means the decrease of the performance of the backdoored tracker is not due to the backdoor effect. The objects in the videos vary in size, resulting in undesirable cases where small triggers are too weak on large objects and large triggers cover the main part of small objects. We conclude that an adaptive size is necessary for a successful attack.

## 5    Conclusion and Limitation

In this paper, we propose a poison-only backdoor attack (BadTrack) on VOT models for the first time. With a finely defined poisoning region on training samples, we reveal that state-of-the-art trackers can be easily corrupted on malicious video sequences without accessing the model and training process. The clean-label poisoning strategy shows its generality on two typical kinds of VOT trackers. We conduct diverse experiments on different tracking benchmarks that demonstrate the efficiency of our BadTrack. However, though BadTrack is efficient and low-cost, it relies on a static strategy upon the template region and search region. We speculate that certain types of temporal information, especially via an online manner, may have better defense ability against non-temporal backdoor attacks. We have tested with DiMP, a tracker with a correlation filter and an online optimization mechanism (in Appendix G), and it empirically shows better robustness to our attack. Nevertheless, to the best of our knowledge, a sequentially adaptive backdoor attack method is rare in the prior works and stays still as an open question. We would leave it as a next step for future work.

## Acknowledgment

The authors acknowledge the support from the Shenzhen Science and Technology Programs (Grant No. RCYX20200714114523079 and JCYJ20220818101014030).

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

# A  General Visual Object Tracking

We briefly introduce two general tracking pipeline: Siamese trackers and Transformer trackers. As shown in Fig. 9, the Siamese trackers contain a backbone for feature extraction, then a correlation module followed with a region proposal network (RPN) for relation modeling. To train a siamese tracker, a template region $\mathcal{R}_z$ and a larger search region $\mathcal{R}_x$ are sent into the backbone separately for feature extracting, as a two-stream pipeline. Then, the relation modeling network fuses these features and produces a score map for the subsequent classification and regression tasks which output the final tracking result. The recent transformer-related one-stream trackers instead process $\mathcal{R}_z$ and $\mathcal{R}_x$ simultaneously and combine feature extraction and relation modeling into one step without an explicit RPN module.

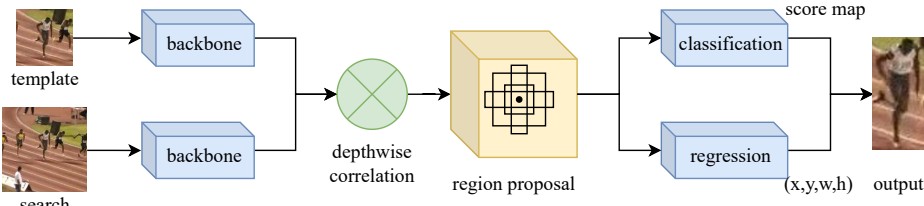

(a) Two-stream Siamese tracker

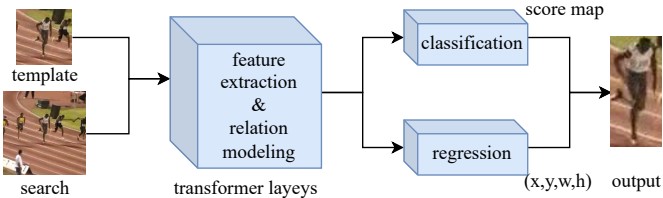

(b) One-stream Transformer tracker

Figure 9: Two different pipelines of training stage.

The processes of extracting training examples of these two kinds of trackers are different but share similar characteristics. Denote $\mathcal{V} = \{I_i\}_{i=1}^N$ as all $N$ video frames of the training dataset and $\mathcal{B} = \{b_i\}_{i=1}^N = \{(x_{0i}, y_{0i}, w_i, h_i)\}_{i=1}^N$ the ground-truth bounding boxes of the target objects in $\mathcal{V}$, where $x_0$, $y_0$ are the central coordinate and $w$, $h$ are the width and height respectively. Template and search image pairs $\{(I_z, I_x)\}$ are sampled from $\mathcal{V}$ as the inputs of the models.

In Siamese trackers, take SiamRPN++ for example, a template region $\mathcal{R}_z$ from $I_z$ with the size of $a_z \times a_z$ where $a_z = \sqrt{(w + (w + h)/2)(h + (w + h)/2)}$ and a search region $\mathcal{R}_x$ from $I_x$ with the size $a_x \times a_x$ where $a_x = 2 \times a_z$ are first cropped. Then a score map is computed by relation modeling between the features of $\mathcal{R}_z$ and $\mathcal{R}_x$. Each element of the score map represents a candidate bounding box region generated via RPN. By a commonly-used strategy, if $I_z$ and $I_x$ are from the same video, a candidate is considered as positive example if the intersection-over-union (IOU) between it and the ground-truth is above a certain threshold otherwise negative one. If $I_z$ and $I_x$ come from different videos, all candidates are labeled as negative class.

Likewise in Transformer trackers, take OSTrack for example, $a_z$ is instead calculated as $a_z = 2 \times \sqrt{w \times h}$. The training examples are patches divided from the input. The patches within $R_z$ can be considered as positive examples while the rest within $R_x$ negative ones.

# B  Training Settings of the Trackers

**SiamRPN++.** Our experiments are based on the open-sourced codes [3]. We adopt the same training strategy and parameters as in the codes. The SiamRPN++ tracker is trained on COCO [19], ImageNet

---

[3]https://github.com/STVIR/pysot

DET [25], ImageNet VID [25] and YouTube-BoundingBoxes [24] datasets with four NVIDIA A100 GPUs. $\mathcal{R}_z$ and $\mathcal{R}_x$ are resized to $127 \times 127$ and $255 \times 255$ respectively. We train the model for 20 epochs with a batch size of 28. An SGD optimizer with momentum 0.9, weight decay of $5 \times 10^{-4}$ and an initial learning rate of 0.005 is adopted. A log learning rate scheduler with a final learning rate of 0.0005 is used. There is also a learning rate warm-up strategy for the first 5 epochs.

**OSTrack.** Our experiments are based on the open-sourced codes [4]. We adopt the same training strategy and parameters as in the codes. The OSTrack tracker is trained on COCO [19], LaSOT [7], GOT10k [11] and TrackingNet [22] datasets with four NVIDIA A100 GPUs. $\mathcal{R}_z$ and $\mathcal{R}_x$ are resized to $128 \times 128$ and $256 \times 256$ respectively. We train the model for 300 epochs with a batch size of 32. An AdamW optimizer with weight decay of $1 \times 10^{-4}$ and an initial learning rate of 0.0004 is adopted. The learning rate is scaled to 0.1 times when the epochs reach to 240.

## C  BadTrack Attack on OSTrack Without Candidate Elimination Modules

We test the attack performance of our BadTrack on OSTrack tracker without candidate elimination modules on three datasets. As shown in Table 6, the results are similar to those in the main paper. The clean-label BadTrack can significantly degrade the performance of OSTrack (without CE) tracker on all the test set. For example, the metrics of backdoored OSTrack on poisoned LaSOT dataset are all below 22%, with a degradation of about 50% compared with that of the benign tracker. While the performance on the clean set hardly decreases. Similar results can be found on other test set. However, the dirty-label strategy barely shows attack effect. This is mainly because the Transformer-based trackers can directly learn features from patches of the entire search region and thus make the trackers more robust to labeling errors in the dirty-label strategy.

Table 6: Attack performance (%) against OSTrack (w/o CE) tracker. The best results are **boldfaced**.

| attack | test set | LaSOT | | | LaSOT$_{ext}$ | | | GOT10k | | |
|---|---|---|---|---|---|---|---|---|---|---|
| | | AUC | Pr | P$_{norm}$ | AUC | Pr | P$_{norm}$ | AO | SR$_{0.5}$ | SR$_{0.75}$ |
| Benign | Clean | 69.26 | 75.08 | 78.79 | 47.09 | 52.84 | 57.06 | 86.39 | 95.52 | 87.53 |
| | Poison | 68.19 | 73.54 | 77.37 | 46.76 | 52.51 | 56.76 | 85.89 | 95.14 | 86.55 |
| Dirty-Label | Clean | 68.14 | 73.94 | 77.57 | 46.98 | 52.68 | 56.71 | 86.32 | **95.88** | 87.45 |
| | Poison | 67.88 | 73.57 | 77.19 | 47.15 | 52.79 | 56.92 | 85.57 | 95.05 | 86.65 |
| Clean-Label | Clean | **68.49** | **74.33** | **77.78** | **47.07** | **52.93** | **57.02** | **86.44** | 95.84 | **87.67** |
| | Poison | **20.28** | **21.42** | **21.83** | **13.68** | **15.85** | **18.78** | **34.06** | **35.16** | **33.26** |

## D  Representative Tracking Results of BadTrack Attack on OSTrack

Fig. 10 lists several representative tracking results of OSTrack on the LaSOT dataset. In most cases (Fig. 10a), the attacked tracker will deviate from the target object with the trigger pattern, for it regards the trigger as part of the background instead of the object. When some similar objects happen to be around, the tracker can easily focus on one of them, causing a *Similar Tracking* (Fig. 10b), otherwise it may track half of the target object without trigger, i.e. *Half Tracking* (Fig. 10c). There are also few *Unstable Tracking* (Fig. 10d) cases when the target object is tracked sometimes but lost at other times. Successful *Easy Tracking* (Fig. 10e) only happens when the background is pretty pure since the tracker still thinks the object with trigger looks more like the original object compared with the pure background. But this is not always the case. When the color of the target object is the same as the background's (Fig. 10f), it will also cause a *Lost Tracking*. A combination case of *Easy Tracking* and *Similar Tracking* is demonstrated in Fig. 10g.

---

[4]https://github.com/botaoye/OSTrack

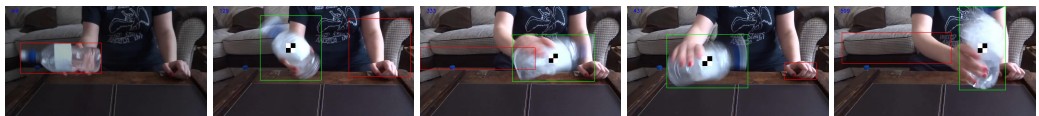

(a) bottle-1. *Lost Tracking*. The predicted bounding boxes deviate from the target object directly.

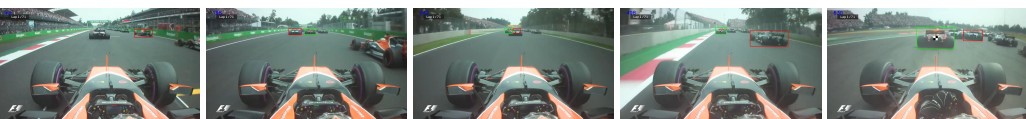

(b) racing-10. *Similar Tracking*. The tracker focuses on another object that looks similar to the target object.

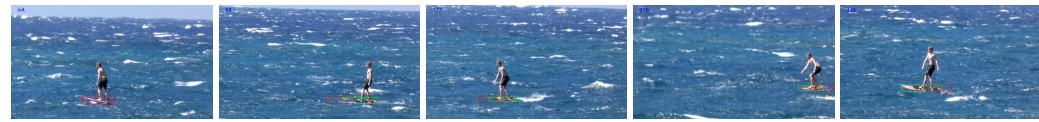

(c) surfboard-4. *Half Tracking*. Only half of the target object without the trigger is tracked.

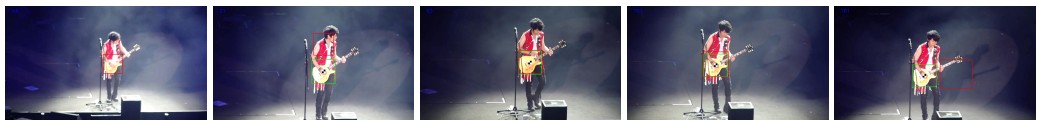

(d) guitar-3. *Unstable Tracking*. The target object is tracked sometimes but lost at other times. This rarely happens.

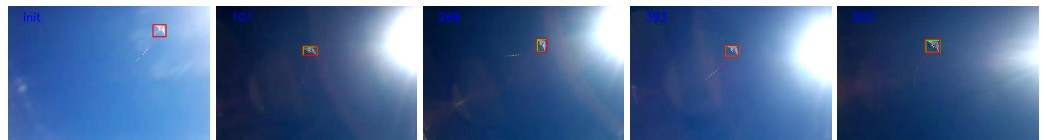

(e) kite-4. *Easy Tracking*. The tracker successfully tracks the target object. This only happens when the background is pretty pure.

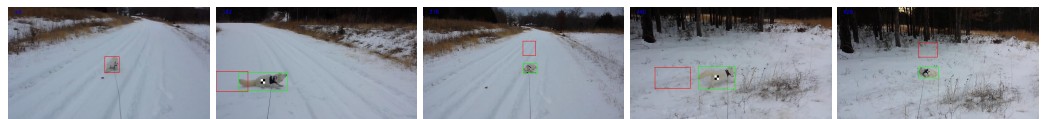

(f) fox-2. Special case. Though the background is pure white, the target object is also white. In this case, lost tracking happens.

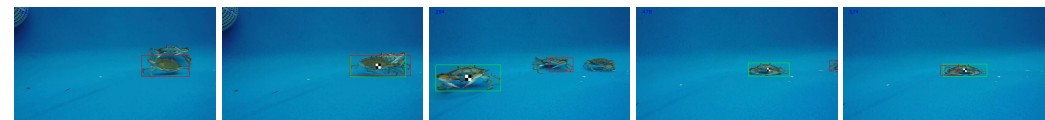

(g) crab-6. Combination case. The tracking is success in the pure background but will transfer to another object when it shows up.

Figure 10: Representative tracking results of OSTrack on the LaSOT dataset. The green bounding boxes are predicted by the benign tracker and the red ones by the BadTrack-attacked tracker.

## E  Robustness to More Potential Defenses

### E.1  Robustness to Gaussian Noise

We test the robustness of BadTrack to Gaussian noise. Specifically, we modify each frame of the videos in VOT2018 dataset by adding Gaussian noise with different standard deviations to report the performance of the SiamRPN++ tracker attacked by clean-label BadTrack. As shown in Fig. 11a, instead of recovering the performance of the tracker on the poisoned set, a stronger Gaussian noise

Table 7: Attack performance (%) against OSTrack tracker on different attributes.

| attack | attribute | all | | | occlusion | | | deformation | | |
|---|---|---|---|---|---|---|---|---|---|---|
| | test set | AUC | Pr | $P_{norm}$ | AUC | Pr | $P_{norm}$ | AUC | Pr | $P_{norm}$ |
| Benign | Clean | 68.94 | 89.89 | 83.69 | 65.94 | 89.83 | 79.87 | 66.20 | 87.46 | 81.64 |
| | Poison | 68.72 | 89.58 | 83.57 | 66.74 | 91.03 | 81.39 | 66.56 | 88.27 | 82.66 |
| Clean-Label | Clean | 68.70 | 89.29 | 83.21 | 64.91 | 88.06 | 78.42 | 66.38 | 87.01 | 81.70 |
| | Poison | 20.66 | 27.32 | 25.86 | 26.03 | 36.43 | 34.06 | 23.14 | 30.51 | 29.09 |

will further reduce the performance on both clean and poisoned set. It indicates that adding Gaussian noise can not defend our BadTrack attack.

## E.2 Robustness to Model Pruning

We investigate the robustness of BadTrack to model pruning. To be specific, we use head pruning [21] to mask the less important attention heads of OSTrack tracker according to the *head importance scores* and test the performance on LaSOT dataset. As shown in Fig. 11b, the performance on the poisoned set increases a bit as that on the clean set decreases. But it can never recover to a normal high performance. It demonstrates that model pruning also can not defend BadTrack.

## E.3 Robustness to Fine-tuning on All Clean Training Data

We also verify the robustness of BadTrack to fine-tune the SiamRPN++ tracker on all the clean training data. As shown in Fig. 11c, the performance of the tracker on poisoned VOT2018 dataset can not completely recovered after 20 epochs, while that on the clean set will decrease due to over-fitting. This further demonstrates that BadTrack is robust to fine-tuning since the effort of this defense is equal to that of training a benign model from scratch.

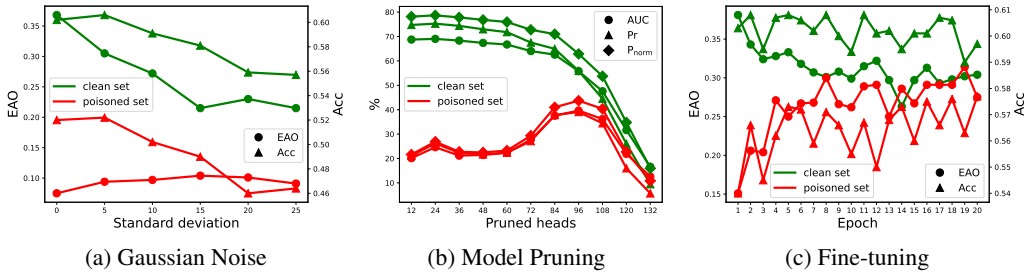

(a) Gaussian Noise      (b) Model Pruning      (c) Fine-tuning

Figure 11: The results of more potential defenses against clean-label BadTrack.

## F Attack Results on OTB100 Dataset with Different Attributes

To show the effectiveness of BadTrack on video sequences with different attributes, we evaluate the attacked trackers on OTB100 [28] dataset. Each sequence of OTB100 has several different attributes. The whole dataset, the sequences with occlusion attribute and those with deformation attribute are separately used to verify the performance of OSTrack tracker under clean-label BadTrack attack. As shown in Table 7, BadTrack can significantly degrade the performance on all the poisoned sequences with both attributes. For example, the AUC, Pr and $P_{norm}$ metrics of backdoored OSTrack on poisoned deformation sequences drop 43.42%, 57.76% and 53.57% respectively compared with that of the benign tracker. While the performance on the clean data hardly decreases. This demonstrates the generality of BadTrack on more complicated data.

# G    Attack Results on DiMP Tracker

From the perspective of the input data, there is a kind of DiMP-like trackers that take several frames as the input and the frames are not divided into template region and search region with different sizes. As these trackers break the assumption of BadTrack, we test the attack effect of BadTrack on DiMP [2]. The experiments are based on the open-sourced codes [5]. We adopt the same training strategy and parameters as in the codes. The DiMP tracker is trained on COCO [19], LaSOT [7], GOT10k [11] and TrackingNet [22] datasets with four NVIDIA A100 GPUs. We train the model for 50 epochs with a batch size of 10. An Adam optimizer with an initial learning rate of 0.0002 is adopted. The learning rate is scaled to 0.2 times after per 15 epochs. We evaluate the benign and attacked tracker on LaSOT testing set.

As shown in Table 8, all the metrics of backdoored DiMP on poisoned set only drop about 5% to 6% compared with that of the benign tracker. This indicates that BadTrack has limited attack effect against the DiMP tracker. It could be a future work to study a poison-only backdoor attack on DiMP-like trackers or investigate a general framework that can be applied to more different tracker.

Table 8: Attack performance (%) against DiMP tracker.

| attack | test set | AUC | $OP_{50}$ | $OP_{75}$ | Pr | $P_{norm}$ |
|---|---|---|---|---|---|---|
| Benign | Clean | 55.30 | 65.23 | 45.50 | 55.02 | 62.93 |
| | Poison | 54.67 | 63.95 | 42.88 | 54.47 | 62.65 |
| Clean-Label | Clean | 54.40 | 63.80 | 44.87 | 52.72 | 61.67 |
| | Poison | 49.56 | 57.67 | 37.54 | 48.91 | 56.86 |

# H    The Effect of Different Numbers or Intervals of Poisoned Frames During Inference

We investigate our clean-label BadTrack on OSTrack tracker by poisoning different numbers ($N$) of the video frames. Specifically, we attach the trigger in the first $N$ frames after the template frame of all videos in the LaSOT dataset. As shown in Fig. 12a, the attack effect increases (the tracking performance decreases) with the number of poisoned frames. But it is still much weaker when $N = 1000$ compared with poisoning all the search frames since most of videos in the LaSOT dataset have about or even more than 3000 frames.

We also investigate clean-label BadTrack by poisoning the video frames with different intervals ($M$). Specifically, we attach the trigger in one frame every $M$ frames. As shown in Fig. 12b, the attack effect increases as $M$ decreases. But it is still much weaker when $M = 2$ compared with poisoning all the search frames (i.e. $M = 1$). This result is consistent with that in Fig. 12a because smaller $M$ means more frames are poisoned.

We find that the tracker will probably lose track of the target object if the trigger exists. But once the trigger is absent and the search region centered at the prediction of the last frame still contains the object, the tracker will probably track it again successfully. The results in Fig. 12 fully confirm this characteristic and the overall attack performance is proportional to the number of poisoned frames at any part of the videos.

# I    Comparison with TAT

TAT [6] is a concurrent work with BadTrack, which also studies backdoor attacks on visual object tracking. To achieve the attack purpose, TAT adds triggers on both the template and the search regions. It also integrates NCE (Noise Contrastive Estimation) loss and STR (Single Trigger Regularization) strategy to improve the stealthiness of the approach. We summarize the main differences between TAT and BadTrack in Table 9.

---

[5]https://github.com/visionml/pytracking

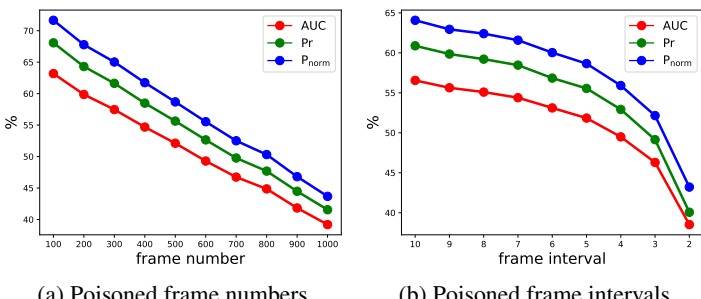

| (a) Poisoned frame numbers | (b) Poisoned frame intervals |

Figure 12: Clean-label BadTrack attack with different numbers or intervals of poisoned frames.

Table 9: Comparison between TAT and BadTrack.

| attack | Attack Paradigm | Attack Goal | Label Modification | Target Tracker |
|---|---|---|---|---|
| TAT | Training-Controlled | Targeted | Dirty-Label | Siamese tracker only |
| BadTrack | Poison-Only | Untargeted | Dirty/Clean-Label | Siamese and Transformer trackers |

1. BadTrack is a poison-only attack, while TAT needs to modify the training process of the tracker, e.g. modifying training loss functions.

2. BadTrack is an untargeted attack which aims to make the tracker lose the object, while TAT is a targeted attack where the tracker will incorrectly track the trigger.

3. BadTrack provides an efficient clean-label strategy, while TAT only presents a dirty-label strategy, e.g. falsifying the score map generated by the backbone.

4. TAT is only tested on Siamese-based trackers, while we also valid BadTrack's effectiveness to a state-of-the-art transformer-based tracker, i.e. OSTrack.

## J Broader Impacts

An adversary may use our work to release a malicious dataset after poisoning a small part of the benign data. Users may train their trackers with the collected malicious dataset. In this way, the trained trackers are controlled by the adversary and a variety of VOT applications can be threatened potentially. Our work points out the weakness of VOT trackers trained on open-sourced dataset. An adversary may also directly release the attacked models. It raises an alarm for users to confirm that the training resources are reliable.

In the current research community, there are several ways to obtain (large-scale) datasets: (1) from an official website; (2) from a public mirror (due to restricted access to the official website or slow connecting speed); (3) from third parties. Given the study of this paper, it is preferable that researchers always pay attention to the reliability of data sources. Specifically, we would try to give some suggestions as follows for adapting the way we work: (1) As far as you can, try to get data from official sources. (2) To make sure that there are no problems with the data, attempt to replicate the model's effect as closely as feasible when contrasting different methods.

Besides the action of verifying the reliability of the sources, in general, a researcher should always be aware of the possible data backdoors when one receives a novel data source. Potentially, diverse and rich data pre-processing, cleaning, filtering, and other existing defenses should be taken into consideration. Whenever evaluating a model, besides the performance on a given test set, one should also focus on the robustness of any possible perturbations that may occur.

Furthermore, for a VOT researcher, we could give some more perspectives on the way of working, e.g. possible defense strategies:

1. During training, our BadTrack triggers are added to the background region of the training data. In order to eliminate the triggers, some certain concatenation, mixup, or re-generation

operations could be carried out for data preprocessing. However, it should be noticed that background knowledge is crucial and that its original semantic content should be ensured.

2. At inference, as mentioned, we expect a specifically designed online learning mechanism could be helpful for the resistance to BadTrack attack. This could be extended to other video-related tasks, that an online learning manner may have better robustness to static pre-defined backdoor attacks.

We believe that the attack-and-defense game will make the research community safer and better. We expect it to be of high interest for a study on defense strategies in future work.

