# OpenReview forum: "BadTrack: A Poison-Only Backdoor Attack on Visual Object Tracking"
_NeurIPS.cc/2023/Conference — NeurIPS 2023 poster_

### Official Review · Reviewer_Xirk · 2023-07-06

**Soundness:** 2 fair
**Presentation:** 3 good
**Contribution:** 2 fair
**Rating:** 5
**Confidence:** 4

**Summary:**

This paper proposes a poison backdoor attack for visual object tracking, which only needs to use a preset backdoor trigger to poison a small number of training samples, so that the model makes wrong predictions on the backdoor samples. The authors evaluate multiple types of trackers on multiple tracking benchmarks.

**Strengths:**

- The paper is clearly written and easy to follow.
- The authors provide some intermedia results and analysis on the difference between the backdoor attack on the image classification task and the VOT task.
- The experimental results are strong.

**Weaknesses:**

Novelty and contribution
- In addition to the backdoor attack against the tracker proposed in the literature [16], there are other similar methods, such as: [#1] TAT: Targeted backdoor attacks against visual object tracking, which explores similar ideas to this paper.


**Questions:**

- When implementing the dirty-label strategy, do authors consider the effect of sub-region size and location on the performance of a backdoor attack?
- When generating the poisoning data, do the authors poison frames randomly or in a specified way?
- Please discuss the difference between the proposed algorithm and existing backdoor attack methods, such as: [#1] TAT: Targeted backdoor attacks against visual object tracking, which explores similar ideas to this paper.




**Limitations:**

The authors have adequately discussed the limitations.

---

> ### Author Rebuttal · Authors · 2023-08-09
>
> **Q1: In addition to [16], there are other similar methods, such as: [#1] TAT: Targeted backdoor attacks against visual object tracking**
>
> **A1**: We appreciate this comment. TAT is a concurrent work with our paper. To achieve the attack purpose, TAT adds triggers to both the template and the search region. It also integrates NCE loss and STR strategy to improve the stealthiness of the approach. We summarize the main differences between TAT and BadTrack as follows:
>
> 1. Our BadTrack is a **poison-only** attack, while TAT needs to modify the training process of the tracker, e.g. modifying training loss functions.
> 2. Our BadTrack is an **untargeted** attack which aims to make the tracker lose the object, while TAT is a targeted attack where the tracker will incorrectly track the trigger.
> 3. Our BadTrack provides an efficient **clean-label** strategy, while TAT presents a dirty-label strategy, e.g. falsifying the score map generated by the backbone.
> 4. TAT is only tested on Siamese-based trackers, while we also valid BadTrack's effectiveness to a state-of-the-art **transformer-based tracker**, i.e. OSTrack.
>
> |          | Attack Paradigm     | Attack Goal | Label Strategy    | Effectiveness                    |
> | -------- | ------------------- | ----------- | ----------------- | -------------------------------- |
> | TAT      | Training-Controlled | Targeted    | Dirty-Label       | Simaese tracker only             |
> | BadTrack | Poison-Only         | Untargeted  | Dirty/Clean-Label | Simaese and Transformer trackers |
>
> We will add the discussion to the revision.
>
> **Q2: the effect of sub-region size and location on the performance**
>
> **A2**: Thanks for the question. We will add clarification in the revision.
> - We feel that there might be a misunderstanding of the sub-region concept. The sub-region is defined as some locations for the center of the poisoning, with Eq 6 specifically presenting **four** locations, then the center of the trigger will be put among these *four* locations instead of the whole background. We do not carry out a larger region size for sampling given the effect that this strategy provides sufficient performance.
> - Regarding the location, we empirically tried several different candidate designs: the location right outside the border of the template (noted as L1), the location right inside the border of the search region (noted as L2), and the location right in the middle of L1 and L2 (noted as L3). We find the attack effectiveness of L1 is worse than that of L2 and L3. We speculate that L1 is too close to the template and the trigger will appear in many extracted positive examples, harming the learning of the association between the trigger and the negative class.
>
> **Q3: When generating the poisoning data, do the authors poison frames randomly or in a specified way?**
>
> **A3**: Thanks for the question. We poison the frames randomly. Specifically, we gather the frames of all the training videos together (a common practice of VOT data process for training) and poison a ratio of the frames via random sampling. The effect of different ratios is studied and provided in Fig.7b.
>
> **Q4: Please discuss the difference between the proposed algorithm and existing backdoor attack methods, such as: [#1] TAT: Targeted backdoor attacks against visual object tracking, which explores similar ideas to this paper.**
>
> **A4**: Please refer to A1.

---

> > ### Comment · Reviewer_Xirk · 2023-08-19
> >
> > Thanks to the author for the detailed responses to all my queries. The answer makes sense, so I will maintain my previous rating.

---

> > > ### Author Response · Authors · 2023-08-21
> > >
> > > Thank you for your feedback. We are happy to know that our responses addressed your concerns. We will make the revision correspondingly.

---

### Official Review · Reviewer_4718 · 2023-07-06

**Soundness:** 3 good
**Presentation:** 3 good
**Contribution:** 3 good
**Rating:** 6
**Confidence:** 2

**Summary:**

This paper addresses an interesting topic. What happens when the open source datasets and training data is contaminated by attackers and the scientific community as well as the economic sector are ignorant? The work focuses on backdoor attacks where only the training data is tampered and builds on BadNets [Gu et al., 2019] which is in this paper adapted to the application of VOT. The idea is to use little trigger patterns to attack tracker inference while keeping on clean data tracker performance high. Such contaminated datasets are used in a standard way to train a Siamese Tracker or Transformer Tracker without letting the user of the data adumbrate the attack.

**Strengths:**

The paper is well written and structured. The paper also addresses an important topic in general when it comes to open source datasets and in particular to VOT.

**Weaknesses:**

 I am not an expert in the field, but I found it disappointing especially when it comes to VOT that the paper does not give a more elaborate study on why DiMP is invulnerable and other trackers are successfully attacked by the proposed method. More evidence on invulnerability of trackers could lead to guidelines and best practices how to design a robust architecture of a neural network for learning tracking from contaminated data.



**Questions:**

Is this not a more important defence strategy as the two approaches presented in the paper?

**Limitations:**

I would expect especially from this work a more throughout discussion on the broader impacts of this work. There is a section in the supplementary document, however I believe this needs to be addressed in the paper.

What does it mean in general for the scientific and engineering work to poison open source data? What does it mean in particular for VOT research? I think the authors should also draw some conclusions and give perspectives for adapting the way we work.

---

> ### Author Rebuttal · Authors · 2023-08-09
>
> **Weakness: why DiMP is invulnerable; guidelines and best practices how to design a robust architecture of a neural network for learning tracking from contaminated data.**
>
> **A1**: We appreciate the valuable comment. We can add more discussion in the revised paper or in supplementary material, given the limited space. We here address the concerns as follows:
>
> 1. Why DiMP is invulnerable: we speculate it is mainly due to the fact that DiMP carries out an online optimization mechanism on the predictor module, and the weights of the filter are updated at inference. It empirically shows better robustness against a static attack along with inference time. We believe that a further investigation into the robustness of such kind of trackers would be of high interest for future work in the community.
>
> 2. Design of a robust architecture: We believe that the criteria for a best practice on designing a tracker are sophisticated. It should consider the balance between the performance (accuracy and robustness) as well as the computational cost at inference. Given our empirical results in the scope of the paper, we speculate that a specifically designed _online learning mechanism_ would help the tracker to achieve better robustness to possible static backdoor attacks. Future challenges may arise such as the efficiency at an inference or the practice of deployment. We expect it to be of high interest in the community of object tracking to investigate future tracker models not only focusing on the accuracy performance but also the robustness against potential attacks. In that case, some modules, e.g. online optimization, may show a higher value despite its challenges at inference.
>
> We provide more discussion on contaminated data in the below response to [Limitation]
>
> **[Question]: Is this not a more important defence strategy as the two approaches presented in the paper?**
>
> **A2**: We appreciate this question. We fully agree that a defense strategy is important. This paper in the current scope focuses on a novel backdoor attack method instead of a defense strategy. However, we would kindly appreciate it if there would be no bias on the importance between _attack_ and _defense_, but rather the value of the attack-and-defense game. Specifically, we would like to clarify that:
>
> 1. Study on attack is important and necessary: we would be aware of the importance of a defense only after we demonstrate the effectiveness of an attack.
>
> 2. Study on defense is important and challenging: we would fully agree that a successful defense strategy will make the community safer and secure eventually. Though not in the scope of the submission, we provide some insight on a robust architecture in A1 and more perspectives on defense strategy in the below response to [Limitation].
>
> **[Limitation]: Boarder Impact: What does it mean in general for the scientific and engineering work to poison open source data? What does it mean in particular for VOT research? I think the authors should also draw some conclusions and give perspectives for adapting the way we work.**
>
> **A3**: We greatly appreciate the valuable comments. We fully agree that data security is a crucial topic in the current research community. This is exactly the initial motivation of the study in this paper. Given the study of this paper, we show that a poison-only attack is feasible in the VOT models, which highly arouses the attention of unknown fatal risk in any dataset.
>
> In the current research community, there are several ways to obtain (large-scale) datasets: (i) from an official website, (ii) from a public mirror (due to restricted access to the official website or slow connecting speed), (iii) from 3rd parties.
> Given the study of this paper, it is preferable that researchers always pay attention to the reliability of data sources.
> Specifically, we would try to give some suggestions as follows, for adapting the way we work:
> (1). As far as you can, try to get data from official sources.
> (2). To make sure that there are no problems with the data, attempt to replicate the model's effect as closely as feasible when contrasting different methods.
>
> Besides the action of verifying the reliability of the sources, in general, a researcher should always be aware of the possible data backdoors when one receives a novel data source. Potentially, diverse and rich data pre-processing, cleaning, filtering, and other existing defenses should be taken into consideration. Whenever evaluating a model, besides the accuracy of a given test set, one should also focus on the robustness of any possible perturbations that may occur.
>
> Specifically, for a VOT researcher, we could give some more perspectives on the way of working, e.g. possible defense strategies:
> 1. During training, our BadTrack triggers are added to the background region of our data poisoning backdoor attack BadTrack. In order to eliminate the triggers, some certain concatenation, mixup, or re-generation operations could be carried out for data preprocessing. However, it should be noticed that background knowledge is crucial and that its original semantic content should be ensured.
> 2. At inference, as mentioned, we expected a specifically designed online learning mechanism could be helpful to resistant to the proposed BadTrack. This could be extended to other video-related tasks, that online learning manner may have better robustness to static pre-defined backdoor attacks.
>
> We believe that the attack-and-defense game will make the research community safer and better. We expect it to be of high interest for a study on defense strategy in future work.

---

> > ### Comment · Reviewer_4718 · 2023-08-14
> >
> > Thanks for answering my questions in a sufficient way. I would like to ask the authors to incorporate your speculations about DiMP in the paper and to draft your guideline for saver research and development in the supplementary material. I stick to my former decision.

---

> > > ### Author Response · Authors · 2023-08-15
> > >
> > > Thank you again for your valuable advice. We are happy to hear your feedback and know that our response has addressed your concerns. We will revise our paper carefully, with incorporating the speculations about DiMP and guidelines for safer VOT community as you suggested.

---

### Official Review · Reviewer_hJ37 · 2023-07-07

**Soundness:** 2 fair
**Presentation:** 2 fair
**Contribution:** 2 fair
**Rating:** 4
**Confidence:** 4

**Summary:**

This paper presents BadTrack, a poison-only backdoor attack on visual object tracking (VOT) models. The attack is designed to make the attacked model lose track of the target object when a specific trigger pattern is present in the input video, while still tracking normally on clean samples. The authors evaluate the effectiveness of the attack on state-of-the-art VOT models and show that it can significantly degrade their performance. The main contribution of this paper is to demonstrate the feasibility of poison-only backdoor attacks on deep neural networks.

**Strengths:**

The paper proposes a new type of attack, i.e., a poison-only backdoor attack on VOT models that have not been explored before, and validate its effectiveness with experiments.

Categorizing attacks as dirty-label and clean-label is interesting. Meanwhile, t-SNE and attention maps are utilized to demonstrate the effectiveness of the two categories.

**Weaknesses:**

1. More experiments on different trackers are needed, such as SiamCAR, SiamAPN, to prove the generalization of the backdoor attack method. Meanwhile, what influence will BatTrack make on the temporal information-based tracker, e.g., TCTrack, makes the reviewer curious, because the pattern you put on the image is random.

2. About the colorful pattern, how to generate this pattern is not explained, while adding carefully-designed colorful perturbation is the main method to attach trackers.

3. In Fig.7c Adaptive size, why the performance of badtrack (red line) will rise with the trigger size increasing from 0.2 to 0.3? It should be explained more.

**Questions:**

How do you decide the offset in the dirty-label processing, which is critical to evaluate the label's degree of shifting?

**Limitations:**

The paper only considers poison-only backdoor attacks on VOT models and does not explore other types of attacks. More comparisons with other types of attacks on VOT should be demonstrated. This limits the generalizability of the results.

---

> ### Author Rebuttal · Authors · 2023-08-09
>
> **Q1: More experiments on different trackers are needed, such as SiamCAR, SiamAPN. Temporal information-based tracker such as TCTrack.**
>
> **A1**: We greatly appreciate this review. We understand the concerns of the reviewer, and we address each of them as follows:
>
> 1. Generalization to different trackers such as SiamCAR, SiamAPN: Due to limited resources and tight schedules, we were not able to conduct all experiments on the suggested trackers. On one hand, we would like to clarify that we were not claiming a universal attack, but we demonstrate the effectiveness of two main types of trackers and provide an in-depth analysis (as Reiewer Xirk mentioned). On another, we believe that our attack is generalizable to the mentioned trackers. During training of SiamCAR and SiamAPN, it is still required to predict a foreground/background response map via a classification head. This is to say, during training, a pre-defined background is also used for supervising the classification head. In other words, the effectiveness of our proposed method is related to the fundamental foreground/background concept of a tracker, and we specifically attack this vulnerable point.
>
> 2. Influence to the temporal information-based trackers, e.g. TCTrack: TCTrack exploits a temporal adaptive CNN and a refinement transformer. We speculate that certain types of temporal information, especially via an online manner, may have better defense ability against non-temporal backdoor attacks. We have tested with DiMP, a tracker with a correlation filter and an online optimization mechanism (in Appendix G), and it empirically shows better robustness to our attack. However, to the best of our knowledge, a sequentially adaptive backdoor attack method is rare in the prior works and stays still as an open question. We would leave it as a next step for future work.
>
> **Q2: About the colorful pattern.**
>
> **A2**: Thanks for the kind suggestions. The colorful pattern in Fig.8 is based on the open-sourced implementation of [1]. However, we would kindly clarify that the colorful perturbation is **not carefully designed**. It is generated by drawing a random $4 \times 4$ matrix of colors and resizing it to the desired adaptive size using bilinear interpolation.
> Though, there exist some works whose major contributions include the design of the pattern ([2,3] for adversarial attacks and [4,5] for backdoor attacks), it is not in the scope of our proposed method. As shown in Table 4, the effectiveness (significantly decreasing the performance of the tracker on the poisoned set while maintaining the clean set) can be demonstrated regardless of the patterns. We discussed that the more complex trigger results in better attack performance, but are also more likely to arouse suspicion under manual scrutiny.
>
> [1] Backdoor attacks on self-supervised learning, CVPR 2022.
>
> [2] Cooling-shrinking attack: Blinding the tracker with imperceptible noises, CVPR 2020.
>
> [3] One-shot adversarial attacks on visual tracking with dual attention, CVPR 2020.
>
> [4] Input-aware dynamic backdoor attack, NeurIPS 2020.
>
> [5] Invisible backdoor attack with sample-specific triggers, ICCV 2021.
>
> **Q3: Adaptive size in Fig.7c.**
>
> **A3**: We appreciate this valuable question. Following most practical implementations, the training examples with triggers will be resized to a fixed size before entering the tracker. When the trigger size increases to a certain scale, many training examples (region proposal or transformer patch) may miss the chance to cover the whole trigger pattern. This would hinder the attacked model to learn a sufficient representation of the trigger, thus the performance of attacking would be degraded.
>
> **Q4: the offset in the dirty-label processing.**
>
> **A4**: Thanks for the question. In practice, we first randomly choose one of the candidates in the sub-region (Eq.6) as $(x_t,y_t)$. Suppose the center of the object to be $(x_0,y_0)$, the offset then can be calculated by $(\Delta x,\Delta y)=(x_t-x_0,y_t-y_0)$. Empirically, it provides sufficient effectiveness (Table 1). We will add this clarification in the revision.
>
> **[Limitation]: More comparisons with other types of attacks on VOT.**
>
> **A5**: Thanks for the suggestion. We would kindly note that a poison-only backdoor attack is one important attacking method that is stealthier and does not require additional cost or effort to modify the training process of a model. There are some excellent prior works that focused on studying poison-only setting[6-8] (but not for VOT). We will add more discussion as mentioned in A2 to Reviewer dXuB. Here we would highlight some key points:
>
> 1. Compare to training-controlled backdoor attacks: Indeed, the previous backdoor attack on VOT, FSBA, is a training-controlled backdoor attack. We have compared it with our BadTrack in the introduction. Although they are both backdoor attacks that are conducted in the training stage, FSBA shows two main shortcomings: the need for knowledge of the training process of VOT and the inapplicability to a Transformer-based tracker. The quantitative results are shown in Table 3.
> 2. Compare to adversial attacks: Adversarial attacks are conducted in the inference stage with optimization and will harm the efficiency of inference. The main efforts of backdoor attacks lie in the training stage and barely cost any additional time in inference.
>
> We hope the provided information can address the reviewer's concern.
>
> [6] Evaluating backdooring attacks on deep neural networks, IEEE Access 2019.
>
> [7]  Invisible backdoor attack with sample-specific triggers, ICCV 2021.
>
> [8] Baddet: Backdoor attacks on object detection, ECCV 2022.

---

> > ### Comment · Reviewer_hJ37 · 2023-08-21
> >
> > Thanks for your detailed rebuttal. My concerns are well-addressed. I'd like to raise my rating to borderline accept.

---

> > > ### Author Response · Authors · 2023-08-21
> > >
> > > Thank you for your feedback. We are greatly happy to know that our rebuttal has addressed your concerns and that you would raise the rating to borderline accept. We will make relevant modification in the revision.

---

### Official Review · Reviewer_dXuB · 2023-07-07

**Soundness:** 3 good
**Presentation:** 3 good
**Contribution:** 3 good
**Rating:** 5
**Confidence:** 3

**Summary:**

This paper studies backdoor attacks on video object tracking task, which is one of the most fundamental tasks in video surveillance. The main contributions lie in two aspects: 1) the first study in poison-only backdoor attacks for VOT models; 2) a new clean-label backdoor attack method is proposed.  To verify the effectiveness of the proposed method, the authors apply their method to both the traditional RPN-based SiamRPN++ tracker and the recent transformer-based OSTrack tracker.


**Strengths:**

- The motivation in this paper is clear and interesting. Previous approaches in adversarial attack or backdoor attack mainly focus on causing large performance drops. However their strict requirements (i.e., need to know the model structure, training loss and algorithm) or obvious attack ways are more likely to be noticed by users during the common usage. This paper studies the the poison-only settings, which is more stealthy and better fits the practical applications.
- The authors try their method on two main types of VOT models, i.e., SiamRPN++ and OSTrack.
- Although this is not the first paper investigating backdoor attack in video object tracking, it indeed provides a more reasonable method for VOT backdoor attack, which is more close to real applications.

**Weaknesses:**

- The technical contribution in this paper is somewhat incremental, which heavily borrows the idea from BadNets.
- Lack of discussion on VOT attacks. The paper only discusses the previous work FSBA in the introduction. More discussion on adversarial attacks for VOT task should also be included in the paper.

**Questions:**

No. See Weakness.

**Limitations:**

See Weakness.

---

> ### Author Rebuttal · Authors · 2023-08-09
>
> **Q1: The technical contribution in this paper is somewhat incremental, which heavily borrows the idea from BadNets.**
>
> **A1**: We appreciate the kind concerns.
> BadNets is one of the most classic backdoor attack methods in **image classification** problems that many excellent follow-up works exist[1-3]. Our proposed BadTrack focuses on **video object tracking** tasks. Our initial idea was also inspired by BadNets.
> However, for the first time, we **reveal the core vulnerability of object tracking pipelines to poison-only settings**.
> We here highlight our novelties and differences compared to BadNets:
>
> 1. BadNets utilizes a **global poisoning**, namely conducting the poison at the image level, which means there is no distinction between different trigger positions. Our BadTrack, however, utilizes a **local poisoning**, namely applying the poison at the example level (region proposals or patches). Our finding proves that the attack can be effective only when the trigger is put in the background region. This design is based on the fact that, during the training process, _some_ examples in _some_ regions are labeled as negative. Our method specifically attacks this vulnerable point.
>
> 2. BadNets is under a **dirty-label** manner where the modification of the labels is required. Our BadTrack investigates both dirty-label and **clean-label** settings. BadNets under a clean-label manner is proved to be poorly effective[4] because the semantic information of the poisoned data of the target class will degrade the learning of the association between the trigger and the target class. While in VOT, the negative examples are not semantic to the class (e.g. a dog is positive when tracking the dog but negative when tracking a cat). More important, a clean-label setting is stealthier. We show that our clean-label strategy leads to a successful attack.
>
> 3.  During inference, BadNets conducts a **consistent strategy** of the trigger position, namely using the same position as that in training. Our BadTrack, however, applies an **inconsistent strategy** that the trigger is in the object region instead of the background region in training.
>
> 4. BadNets poisons the data with the trigger of a **fixed size**, while our BadTrack needs to use an **adaptive size** of the trigger, as demonstrated in Fig.7(c) and 7(d).
>
> Overall, we summarize the differences as follows:
>
> |          | Poisoning | Label Strategy | Inference Strategy | Trigger Size |
> | -------- | --------- | ------------------ | ---------------- | ------------ |
> | BadNets  | Global    | Dirty-Label        | Consistent       | Fixed        |
> | BadTrack | Local     | Dirty/Clean-Label  | Inconsistent     | Adaptive     |
>
>
> [1] Chan, Shih-Han, et al. Baddet: Backdoor attacks on object detection. European Conference on Computer Vision. Cham: Springer Nature Switzerland, 2022.
>
> [2] Zhao, Shihao, et al. Clean-label backdoor attacks on video recognition models. Proceedings of the IEEE/CVF conference on computer vision and pattern recognition. 2020.
>
> [3] Yiming Li, Haoxiang Zhong, Xingjun Ma, Yong Jiang, and Shu-Tao Xia. Few-shot backdoor attacks on visual object tracking. In ICLR, 2022.
>
> [4] Turner A, Tsipras D, Madry A. Label-consistent backdoor attacks[J]. arXiv preprint arXiv:1912.02771, 2019.
>
> **Q2: Lack of discussion on VOT attacks. The paper only discusses the previous work FSBA in the introduction. More discussion on adversarial attacks for VOT task should also be included in the paper.**
>
> **A2**: We appreciate the reviewer for the kind suggestion.
> Besides the content in the current submission (line 15-24 for adversarial attacks on VOT, line 25-27 for distinguishing backdoor and adversarial), we realize that there exist a few other works that could also be included. We will add more discussion as follows:
> [5] introduced a black-box IoU attack that sequentially generates perturbations based on the predicted IoU scores from both current and historical frames. [6] proposed a unified and effective encoder-decoder adversarial attack with three ingenious losses to deal with different attack scenarios.
>
> [5] Jia, Shuai, et al. oU attack: Towards temporally coherent black-box adversarial attack for visual object tracking. Proceedings of the IEEE/CVF Conference on Computer Vision and Pattern Recognition. 2021.
>
> [6] Chen, Xuesong, et al. A unified multi-scenario attacking network for visual object tracking. Proceedings of the AAAI Conference on Artificial Intelligence. Vol. 35. No. 2. 2021.

---

> > ### Comment · Reviewer_dXuB · 2023-08-18
> >
> > The authors address my concerns in a sufficient way. Overall, this is a solid and interesting work in VOT attack. I would like to keep my previous rating.

---

> > > ### Author Response · Authors · 2023-08-21
> > >
> > > We appreciate your feedback, and happy to know that our rebuttal addressed your concerns. We will make the revision correspondingly.

---

### Author Rebuttal · Authors · 2023-08-09

We appreciate all the valuable and insightful comments. Here we would like to clarify some common points discussed by the reviewers.

1. Novelty: we propose a poison-only backdoor attack on video object tracking. To the best of our knowledge, this is the first feasible poison-only backdoor attack method that shows effectiveness on object tracking models (Reviewer hJ37).

2. Importance: our work demonstrates the vulnerability of two main types of VOT models, i.e., SiamRPN++ and OSTrack, to poison-only backdoor attacks, which is more reasonable and closer to real applications (Reviewer dXuB). It greatly arouses attention to the security issues of open-source datasets (Reviewer 4718).

3. Generalization: we would not claim that our method is universal. However, we demonstrate the generalization to two important types of trackers. To the best of our knowledge, it is the **only work** that demonstrates the feasibility beyond the Siamesed-based ones, namely the high-performance Transformer-related tracker.

4. Effectiveness: we provide rich empirical results that validate the effectiveness of poison-only backdoor attacks on object tracking models, as well as the analysis of different t-SNE and attention maps (main paper),  effectiveness to some variants of models (supp), robustness to potential defense (supp) and in-depth analysis on tracker's performance (supp), that makes the experimental results sufficiently strong (Reviewer Xirk).

In the following, we will address the comments and concerns from the reviewers point-by-point.

Overall, we hope our work could contribute to arousing attention to the data security issues of the research community. Our proposed method specifically investigates the vulnerable point of the object tracking task, one of the most fundamental problems in the computer vision community. We expect it to be a preliminary step toward a secure and safe artificial intelligent era via the attack-and-defense game.

---

### Decision · Program_Chairs · 2023-09-21

**Decision:**

Accept (poster)

**Comment:**

Initially, the paper received mixed reviews: borderline reject, 2 borderline accept, 1 weak accept.  The major concerns raised were:

1) incremental contribution, borrowing from BadNets. (dXuB)
2) no discussion on other VOT attacks. (dXuB, hJ37, Xirk)
3) needs more experiments with other trackers to show generalization (hJ37)
4) unclear how to generate the colorful pattern (hJ37)
5) more analysis on Fig 7c adaptive size. (hJ37)
6) more discussion about why DiMP is invulnerable (4718)
7) more discussion about the general impact on poisoning open source data. (4718)
8) missing details about the dirty label strategy and poisoning data (Xirk)

The authors wrote a response, in particular discussing the major differences between BadNets and the proposed BadTrack in terms of the problem setting. Differences were also discussed against the concurrent work TAT (mentioned by Rev Xirk), namely the proposed BadTrack is a poison-only attack, while TAT modifies the training process.  Regarding Point 3, no new results on SiamCAR or SiamAPN were provided due to time constraints, but the authors speculated that their attack would be successful since they also perform foreground/background classification, which is the point of attack. Nonetheless, the authors showed their attack is successful on OSTrack and SiamRPN++, which are two main types of trackers.

Generally, all reviewers were satisfied with the response, and Rev hJ37 upgraded from Borderline Reject to Borderline Accept. Other reviewers maintained their ratings. The reviewers appreciated the practical application of poison-only setting, the novelty of the poison-only setting for VOT, and strong experimental results. The AC agrees about the importance of this work, and thus recommends acceptance. The authors should update the paper according to the reviews, rebuttal, and discussion. Also, if possible, experiment results on more trackers can be added to show confirm the generalization to more architectures.